# Reflection of near-infrared light confers thermal protection in birds

Iliana Medina[1], Elizabeth Newton[1], Michael R. Kearney[1], Raoul A. Mulder[1], Warren P. Porter[2] & Devi Stuart-Fox[1]

Biologists have focused their attention on the optical functions of light reflected at ultraviolet and human-visible wavelengths. However, most radiant energy in sunlight occurs in 'unseen' near-infrared (NIR) wavelengths. The capacity to reflect solar radiation at NIR wavelengths may enable animals to control heat gain and remain within their critical thermal limits. Here, using a continent-wide phylogenetic analysis of Australian birds, we show that species occupying hot, arid environments reflect more radiant energy in NIR wavelengths than species in thermally benign environments, even when controlling for variation in visible colour. Biophysical models confirm that smaller species gain a greater advantage from high NIR reflectivity in hot, arid environments, reducing water loss from compensatory evaporative cooling by up to 2% body mass per hour. These results highlight the importance of NIR reflectivity for thermal protection, which may become increasingly critical as the frequency of extreme climatic events increases.

[1] School of BioSciences, The University of Melbourne, Parkville, VIC 3010, Australia. [2] Department of Zoology, The University of Wisconsin Madison, Madison, WI 53706, USA. These authors contributed equally: Iliana Medina, Elizabeth Newton. Correspondence and requests for materials should be addressed to I.M. (email: iliana.medina@unimelb.edu.au)

Sunlight is the most important source of radiant energy for life on earth. About 45% of the energy in sunlight falls within ultraviolet (UV; 300–400 nm) and human-visible (400–700 nm) wavelengths, which are reflected and scattered by surfaces to produce the extraordinary diversity of colours we see in nature. The other 55% of energy in sunlight falls within the near-infrared wavelengths (NIR; 700–~2500 nm), which are not visually perceived by most terrestrial animals due to physical constraints imposed by the chemistry of photoreceptors[1]. Although some animals, such as some snakes, beetles, ticks and mites, have long-wavelength thermal infrared receptors on other parts of the body to sense heat[2,3], these are largely insensitive to NIR light[4]. Given that NIR reflectance (at least >750 nm) cannot be seen, NIR variation is unlikely to be directly influenced by selection for optical functions such as camouflage or communication. Instead, variation in NIR could primarily reflect selection for thermoregulation[5,6].

We tested whether NIR reflectivity (controlling for UV and visible reflectance) varies in relation to thermal environment in a continent-wide analysis of 90 species (12%) of Australian birds (spanning 35% of the families and 66% of the orders in Australia; $n = 616$ individuals) from all major habitat types, including sea and shore birds, waterbirds, forest or arid specialists and habitat generalists (Supplementary Data 1). Around 70% of the Australian continent is characterised by hot, dry environments where it is the upper rather than lower critical thermal limits that drive thermal adaptations[7]. Because of asymmetry in temperature–fitness curves, fitness (including probability of survival) drops

more rapidly at temperatures above than below the optimum[8]. Birds are particularly vulnerable to extreme heat and dryness and should be under strong selection for heat control strategies[9,10]. This is because birds have relatively small body sizes, high internal temperatures, use thermally buffered microsites less than ectotherms and other endotherms, and are commonly diurnal.

We employed phylogenetic comparative methods and biophysical modelling to explore the adaptive significance of NIR reflectivity. Our results show that species in arid and hotter environments have higher NIR reflectivity of body regions exposed to direct sunlight, and that this association was stronger for smaller species. Moreover, smaller species, rather than large ones, may obtain larger thermal benefits from reflectivity in this part of the spectrum.

## Results and Discussion

**Relative NIR reflectivity varies among species.** We found striking variation across species in reflectance across the NIR spectrum (Fig. 1a), even for visibly similar colours. Solar reflectivity (the proportion of incident solar energy reflected) in the UV-visible and NIR wavelength ranges were highly correlated (Pearson's $r > 0.8$, Supplementary Figure 1). Nevertheless, relative NIR (calculated as the residuals from a regression between UV-visible and NIR reflectivity) ranged from very low (mean: −11.66) in the superb fairy-wren (*Malurus cyaneus*) to very high (mean: 6.81) in the azure kingfisher (*Alcedo azurea*), and indicated a maximum of 40% variation in NIR between individuals for a given level of UV-visible reflectivity (Fig. 1a, b,

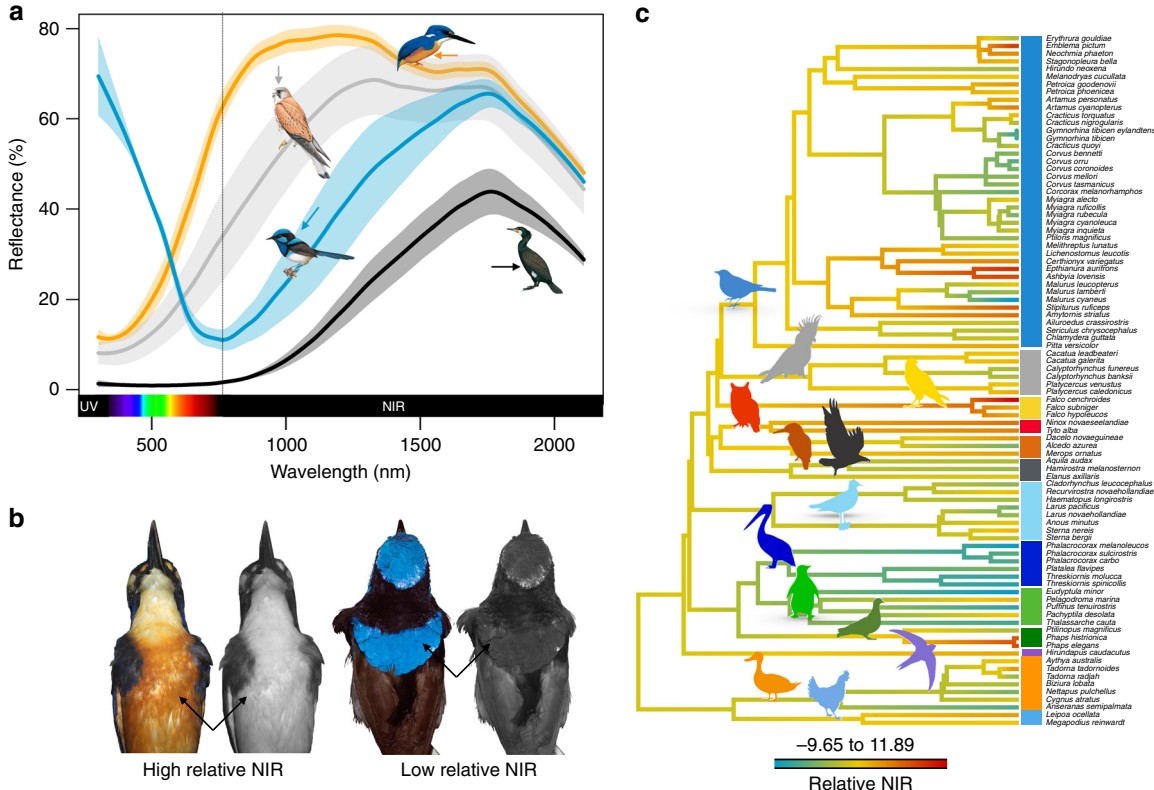

**Fig. 1** Near-infrared (NIR) reflectance variation in Australian birds. **a** Reflectance spectra for representative species with relatively high (light grey and orange) and low (blue and black) NIR reflectivity. Grey: Nankeen kestrel (*Falco cenchroides*, crown); orange: azure kingfisher (*Alcedo azurea*, belly); blue: superb fairy-wren male (*Malurus cyaneus*, mantle); and black: great cormorant (*Phalacrocorax carbo*, belly). Species drawings taken with permission from ref. [31]. **b** Visible (left) and NIR (right) photographs of specimens with high (azure kingfisher) and low (superb fairy-wren) relative NIR. **c** Average relative NIR per species (across dorsal patches) mapped onto a phylogenetic tree (random example from the 1000 trees used in analysis), branch colours represent the value of relative NIR for each species. Vertical bars represent avian order, and correspond to the colour silhouettes on top of the branches

Supplementary Figure 1). Mapping relative NIR reflectivity onto a representative phylogeny of our sampled species (Fig. 1c) reveals moderate phylogenetic conservatism (mean ± s.d., Pagel's $\lambda_{mantle} = 0.44 \pm 0.10$, $\lambda_{crown} = 0.72 \pm 0.23$, $\lambda_{breast} = 0.74 \pm 0.06$, $\lambda_{belly} = 0.92 \pm 0.03$), and substantial variation within and between avian orders. Females tend to have higher values of NIR and UV-visible reflectivity compared to males, but only for ventral plumage patches (Supplementary Figure 2). Shawkey et al.[6] similarly found higher NIR reflectivity in female than in male sunbirds. NIR reflectivity is strongly influenced by the microstructure of feathers, especially the shape of barbs and density of barbules[11]. The difference between males and females in NIR reflectivity of ventral plumage could be due to differences in feather structure related to sex roles in incubation, but this remains speculative.

**NIR reflectivity is associated with thermal environment.** After controlling for the association between NIR and UV-visible reflectivity and for phylogeny, species occupying hotter, drier environments with greater average summer solar irradiation, more extreme temperature days >35 °C and lower annual vapour pressure (higher environmental PC2; Supplementary Table 1, Supplementary Fig. 3) had significantly higher NIR reflectivity of the crown and mantle but not breast and belly (phylogenetically generalised least squares model (PGLS); NIR reflectivity: crown: $t = 3.21$ to $4.91$, df = 85, $P < 0.001$; mantle: $t = 3.16$ to $3.77$, df = 85, $P = 0.001$ to $0.002$; Fig. 2, Supplementary Tables 2, 3). Strikingly, this evolutionary association was much stronger for smaller than for larger species (PC2 × body mass, crown: $t = -4.25$ to $-2.38$, $P = 0.001$ to $0.01$; mantle: $t = -2.89$ to $-2.33$, $P = 0.004$ to $0.02$, Supplementary Tables 2, 3, Fig. 2). The same pattern was found when we used habitat category instead of climate principal components (PCs); arid specialists have

significantly higher relative NIR reflectivity than forest specialists, waterbirds, shore birds or habitat generalists (Supplementary Table 4). In contrast to dorsal plumage, NIR reflectivity of ventral patches was higher in more tropical environments with higher humidity and maximum winter solar irradiation (environmental PC1, Supplementary Table 1; relative NIR: breast: $t = 2.04$ to $4.29$, $P = 0.001$ to $0.03$; belly: $t = 1.06$ to $4.48$, $P = 0.001$ to $0.13$; Supplementary Tables 2, 3, Supplementary Fig. 4). However, this effect was weaker (lower evidence ratios) and less consistent across phylogenetic trees and methods (Supplementary Tables 2, 3; see Methods).

When all patches were combined, results were qualitatively the same as for dorsal patches: NIR reflectivity was higher in hotter, drier environments, particularly for smaller species (PC2 × body mass, all body regions combined: $t = -3.57$ to $-1.86$, $P = 0.0001$ to $0.05$; Supplementary Table 5). Similar overall trends for individual patches were found for males and females (Supplementary Table 6).

Furthermore, we found an association between hotter, drier environments (environmental PC2) and total reflectance (UV-visible + NIR) of dorsal body regions (crown: $t = 1.44$–$2.06$, df = 88, $P = 0.03$–$0.15$; mantle: $t = 2.63$–$2.97$, df = 88, $P = 0.003$–$0.009$; Supplementary Table 7), but not ventral body regions (Supplementary Table 7). Together, these results strongly support the inference that selection for thermal benefits has shaped plumage reflectance properties.

The higher NIR reflectivity of plumage in species occupying hotter, drier environments could be a direct or indirect response to selection for thermal protection. For example, selection for structural feather properties (e.g. an increase in the density of barbules) to reduce penetration of heat through the feathers to the skin could simultaneously increase NIR reflectance. Alternatively,

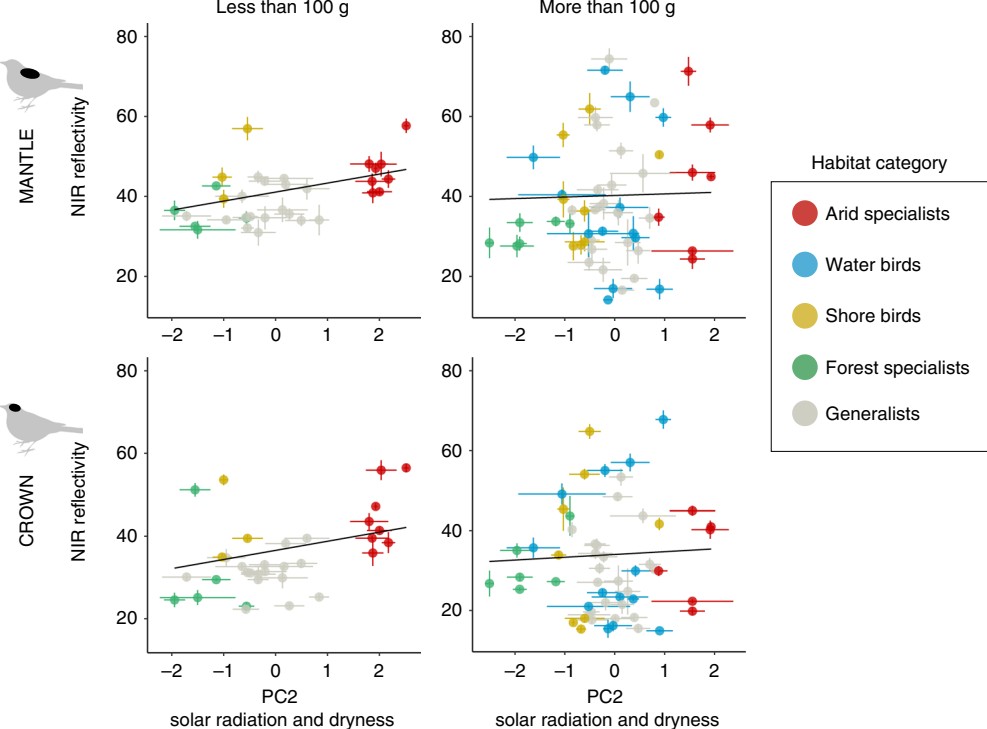

**Fig. 2** Relationship between NIR reflectivity and climate for the mantle and crown. The data are separated by the arbitrary size of 100 g (<100 g: left panels, 33 species; >100 g: right panels, 57 species) to facilitate visualisation of the interaction between environmental PC2 and body mass, but body mass was treated as a continuous variable in statistical analyses. Each point represents the average value per species and the standard error in both axes; colour represents the species' habitat. Trend lines were predicted using a phylogenetically controlled model that included environmental PCs, UV-visible reflectivity and a random phylogenetic tree

dry conditions could influence feather moisture, reducing water absorption bands in the NIR and producing higher NIR reflectance as an indirect side effect[12]. The latter explanation seems less plausible because we would expect feather moisture of all body regions to be similarly affected; yet we found no consistent relationship between NIR reflectivity and arid environments for ventral plumage, and the effect of humidity is opposite for ventral and dorsal patches.

**Biophysical models suggest adaptive benefit**. To assess the potential effects of NIR reflectivity on an individual's fitness in arid environments, we used a biophysical model of an endotherm's energy exchange with its environment[13] integrated with a model predicting microclimates available to the animal from meteorological and environmental data[14,15]. We compared effects of the upper and lower 95% confidence bounds of observed dorsal NIR reflectivity, keeping visible reflectivity constant at the mean, and using specific estimates of parameters for birds of different sizes (10 g, 100 g and 1 kg). High NIR reflectivity results in a maximum reduction in evaporative water loss (EWL) to avoid overheating of 2.12% of body mass per hour for small birds (10 g), 0.24% for 100 g and 0.11% for large birds (1 kg), in arid environments in Australia (Fig. 3a). EWL through the respiratory system and skin as a means of cooling becomes very costly under extreme, prolonged heat events when water is scarce[10,16,17]. Our biophysical model suggests that at temperatures above 40 °C small birds could experience dehydration levels above 10% body mass per hour, reaching a lethal threshold in just a couple of hours[9]. These estimates are higher than those from experimental studies in metabolic chambers where there is no extra load from solar radiation (~5–9% body mass per hour for small birds)[18–21]. However, our model very accurately predicts experimental values from Wolf and Wolsberg[18] under equivalent conditions; i.e. with no solar radiation (Supplementary Figure 5). High NIR plumage reflectivity has the potential to substantially reduce water loss under hot, dry conditions for small birds but has minimal benefit for larger birds (Fig. 3a).

The benefit of higher reflectivity in small birds is possibly due, in part, to a thinner insulating feather layer, and contrasts with the negligible effect of reflectivity on heat load at the skin for large birds[22]. In addition to a thinner insulating layer, smaller animals have a higher surface area to volume ratio and thinner boundary layers, increasing water vapour transport from the animal and coupling them more strongly to convection than to radiative heat exchange[12,23]. However, convection is less effective in reducing heat load when air temperatures are high, and can even increase heat load if environmental temperature exceeds body temperature[23]. Smaller species also are less able to maintain water and fuel reserves to prevent dehydration, and have lower survival during heat waves[10]. If foraging on the ground, they are also exposed to microclimates with lower wind speed and higher air temperatures due to boundary layer effects. Opportunities for behavioural thermoregulation may also be constrained in arid environments, and trade-off with nesting or foraging requirements[10]. Consequently, increased NIR reflectivity may be an important strategy for thermal protection in arid environments, particularly for smaller species.

Environmental parameters in our model represent 'extreme conditions' (upper 95% confidence limit of environmental PC2 in the data set, Simpson Desert Reserve in South Australia [−27.36, 138.71]) but almost half of the Australian continent reaches these extreme conditions during summer. Thirteen per cent of the species sampled and ~10% of the passerine species in Australia (30 species) are distributed in regions with similar aridity, and between 2001 and 2018, more than 3000 records for more than 100 species of passerines were reported in the region of the Simpson Desert (Atlas of Living Australia). Additionally, recent data indicate that the incidence of heat waves and extreme high temperatures will increase[24] and that arid and hyper-arid zones will likely expand[25] suggesting that the likelihood of birds encountering these conditions will probably increase in the future. Furthermore, we found similar potential effects of NIR reflectivity on rates of EWL based on models using environmental data from Arizona, USA (Supplementary Fig. 6), suggesting that the link between NIR reflectivity and environment that we found in Australia has the potential to be a global pattern and could affect a significant number of bird species.

**Trade-offs between visible and NIR reflectivity**. In contrast to the NIR, we found little evidence for climate-associated variation in UV-visible reflectivity, with the exception of a small increase in UV-visible reflectivity of the mantle in hotter, drier environments

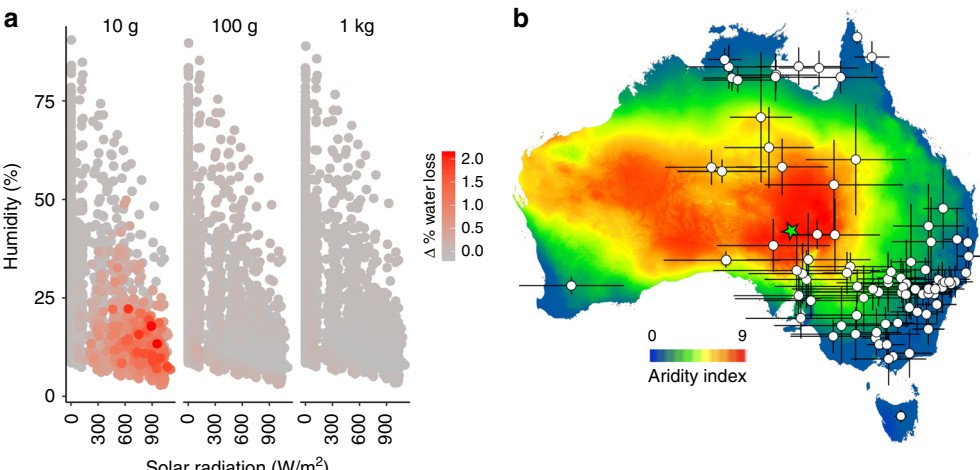

**Fig. 3** Biophysical model results in Australia (Simpson Desert). **a** Estimated reductions in evaporative water loss due to high NIR (40% vs 10% reflectivity) under different combinations of solar radiation and humidity. Higher values (warmer colours) indicate a higher percentage of water saved by having higher plumage reflectivity; grey values (closer to zero) indicate no difference. Values estimated for three different sizes. **b** Aridity index values and mid-distribution points for the 90 species in our data set; bars contain ~50% of 1000 data points. The green star shows the location from which climate forcing data were obtained for the biophysical model. The map was extracted from the Atlas of Living Australia (www.ala.org.au)

(Supplementary Table 3). This is expected, given that the mantle is not only the largest region exposed to direct sunlight but it also has low sexual dimorphism and is likely under weaker selection for alternative functions of coloration (e.g. signalling and camouflage). In many cases, however, there may be a trade-off or compromise between selection for thermal protection and other functions such as camouflage and signalling. For example, in birds with dark or brown plumage, potential thermal costs of high absorptivity (low reflectivity) in the UV-visible spectrum may be offset by benefits of camouflage or protective benefits of melanin against UV damage or abrasion[26]. Equally, for species occupying less thermally hostile environments, selection for optical functions is likely to be the strongest drivers of both visible and correlated NIR reflectivity.

**Conclusions**. Our results suggest that birds, and potentially other animals, may have solved the problem of competing selection for optical and thermal functions by modulating visible and NIR properties differently. Mimicking those solutions in developing artificial materials to enhance energy efficiency holds significant promise[27,28]. Our data also highlight that empirical measures of NIR reflectivity must be incorporated into mechanistic models predicting the effects of climate change, particularly extreme heat events, on individual fitness and species distributions.

## Methods

**Specimen selection**. Specimens were sourced from the ornithological collections of Museums Victoria, the South Australian Museum, and the Australian Museum (Supplementary Data 1). We restricted measurements to geo-referenced, adult specimens with clean, undamaged feathers. Where possible, we selected specimens with information on sex, age, collection year and spanning the full distribution of the species. Between 6 and 10 specimens were measured per species, with an average of 6 measured for sexually monochromatic species (3 female and 3 male) and 10 for dichromatic species (5 female and 5 male).

We measured reflectance (300–2100 nm), encompassing 99% of incident energy in direct sunlight, for 616 individuals belonging to 90 species sampled from 14 orders and 34 families representing the diversity of the Australian avifauna (Fig. 1a; Supplementary Data 1). Species were chosen to represent the full range of environmental and climatic conditions in Australia; including arid, temperate, tropical and coastal species, species with large continent-wide distributions and species with distributions restricted to different climates or latitudes. Species selected included those which feed and nest under complete sun exposure; species that have black plumage despite living in hot, arid zones; species that live under high vegetation cover and are therefore exposed to little solar radiation; and species that have a very wide distribution and experience both extremes of Australian climate.

**Spectral measurements and processing**. Reflectance measurements were taken with a dual-spectrometer system (Ocean Optics, Inc., USA) consisting of two spectrometers (USB2000+ [300–1000 nm] and NIRQuest [1000–2100 nm]) with two light sources (PX-2 pulsed Xenon light for the UV-visible range and HL-2000 tungsten halogen lights for the visible-NIR range) connected with a quadrifurcated 600 µm fibre optic. We used one of two anodised aluminium probe holders depending on the size of the sample or specimen: a block probe holder with an ovoid aperture of 9.5 × 7 mm and a pencil probe holder with an ovoid aperture of 4 × 3 mm. Measurements were calibrated against a diffuse reflectance standard (Spectralon: 97–99% reflectance across the spectral range 300–2100 nm).

We took three measurements from each of four primary areas (patches) on each bird skin: the crown and the mantle (representing the dorsal surface), and the breast and the belly (ventral surface). The probe was angled coincident oblique (45°) to the sample surface and positioned parallel to the direction of the feather, off-centre of the rachis. Measurements were taken towards the tip of pennaceous feathers only. The rachis was only included on feathers that were too small to exclude it from the measurement area of the probe holder. For each patch, we took measurements from the predominant colour of that patch. For example, painted finch ventral measurements were taken on black and white feathers rather than the narrow strip of red. Where two or more colours were equally prominent in a patch (due to coarse pattern or discrete colour patches), we took three measurements within each colour and averaged the resulting spectra to represent that patch. We visually checked each spectrum for measurement anomalies and averaged the three measurements for each patch.

We calculated the proportion of solar energy reflected by feathers, reflectivity, as solar reflectivity, $R = [S(\lambda)I(\lambda)d(\lambda)]/[I(\lambda)d(\lambda)]$, where is reflectance and $I$ is solar irradiance ($S$) across the wavelengths ($\lambda$) of interest. For solar irradiance, we used

the ASTM G-173-03 standard irradiance spectrum for dry air. We calculated reflectivity for the UV-visible spectrum (300–700 nm), NIR (700–2100 nm) and the full spectral range (300–2100 nm).

Due to the strong association between UV-visible and NIR reflectivity ($r^2 > 0.8$; Supplementary Figure 1), we extracted the residuals from a quadratic regression between reflectivity in the UV-visible spectrum and NIR reflectivity. We refer to these residuals as 'relative NIR' (i.e. NIR variation independent of UV-visible reflectivity). We used residuals from a quadratic rather than linear regression because the fit of the former was better (quadratic: $r^2 = 0.89$; linear: $r^2 = 0.85$). Results of phylogenetic comparative analyses (see below) using residuals from a linear regression were qualitatively identical for dorsal patches, but for ventral patches the weak relationship between relative NIR and environmental PC1 was absent when using residuals from a linear regression.

**Predictors: climate and body size**. Climate data were compiled for each specimen locality from continent-wide 0.05° grids of interpolated daily weather data produced by the Australian Water Availability Project (AWAP) database[29]. Data were extracted as average austral summer (December–February), winter (August–July) or annual daily values over a 26-year period (1990–2015 inclusive) because data for solar irradiation are only available from 1990 onwards. Climate variables extracted were maximum and minimum daily summer and winter temperatures, annual vapour pressure, summer and winter solar irradiation, and average annual number of extreme temperature days (days where maximum temperature exceeds 35 °C). Average winter maximum temperature and average summer minimum temperature were highly correlated with average winter solar irradiation ($r = 0.96$ and 0.94) and average winter minimum temperature was strongly correlated with average yearly vapour pressure ($r = 0.94$); whereas, correlations between all other variables were <0.8. Therefore, we used five climatic variables (average maximum summer temperature, average number of days >35 °C, average summer and winter solar irradiation, and average yearly vapour pressure) and summarised these using a PC analysis. The first two PCs accounted for 91% of variation (56% and 35% respectively, Supplementary Table 1) and were used in subsequent analyses. The first PC loaded positively against winter solar irradiation, and to a lesser extent, annual vapour pressure and summer maximum temperature, while PC2 loaded positively against average summer solar irradiation and the number of days >35 °C and negatively against annual vapour pressure (Supplementary Table 1, Supplementary Figure 3). Thus, high values of PC1 represent less seasonal, wetter climates with warm, sunny winters, whereas high values of PC2 represent hot, dry climates.

We classified species into broad habitat categories of waterbirds (species foraging on or in water), sea or shore birds (species foraging on or offshore, exposed to coastal winds and humidity), forest specialists (restricted to wet forests with high vegetation cover), arid specialists (restricted to hot, dry environments) or generalists (those having no or few specific habitat preferences or requirements) based on feeding habitat preference classifications from Garnett et al.[30], field guides and HANZAB[31]. Lastly, species-level (for males and females) body mass data (a thermally relevant measure of body size) were taken from de Hoyo et al.[31] and log10-transformed.

**Phylogenetic comparative analyses**. To test whether variation in either NIR or UV-visible reflectivity could be explained by climate or body mass, we built a PGLS, using information on average colour variables and average climate variables per species. We conducted the analysis at the species level because information on body size was not available for each individual measured. The response variable in the PGLS models was either average absolute or relative NIR reflectivity, or average UV-visible reflectivity per species. For models predicting absolute NIR reflectivity (Supplementary Table 2), the predictors were environmental PC1, PC2, body mass and the interactions between the PCs and body mass, we also included UV-visible reflectivity as a covariate. For models predicting relative NIR or UV-visible reflectivity (Supplementary Table 3), the predictors were environmental PC1, PC2, body mass and the interactions between the PCs and body mass. We ran models using both relative NIR reflectivity (residuals) and absolute NIR reflectivity with UV-visible reflectivity as a covariate to ensure that our results were consistent across methods of accounting for variation in UV-visible reflectivity. Models were run for each body patch separately.

We controlled for phylogenetic non-independence by using the R package 'caper'[32] and running each of the PGLS models described above for 1000 phylogenies obtained from birdtree.org using the Hackett backbone[33]. For each analysis on each tree we used the command 'dredge' in the MuMIn R package[34], which performs automated model selection with subsets of the full model, exploring all possible combination of predictors. For each run we extracted which subset of variables was present in the best model and this process was repeated for each of the 1000 trees. We also extracted information on the evidence ratio between the best and the null model (a measure of how many times better is the best model compared to the null one) and calculated the percentage of trees in which the most common best model was found, when the best model was better than the null model (delta Akaike Information Criterion, AICc > 3 between best and null model[35]).

Finally, for each patch and each response variable (absolute NIR, relative NIR and UV-visible reflectivity) we ran a PGLS with only the predictors present in the most common best model (found in the model selection process described above), across 1000 trees. For each predictor in this model we extracted a 95% highest

posterior density interval for the estimate, *t*-value and the *P*-value across 1000 trees. We report the results of the best models in Supplementary Tables 2 and 3.

**Habitat and sex**. As an independent way of gauging the biological significance of our results, we used habitat category instead of environmental PC2 as a predictor of reflectivity. The model was run in the same way as the PGLS described above, but in this case habitat category and mass were the only predictors. We present the results in Supplementary Table 4.

Given that our sample contained several species that are sexually dichromatic, we repeated the analysis presented in Supplementary Table 3, but we ran it separately for females and males of each species, averaging the reflectivity values and environmental PC values within each sex of each species. We used the same type of analysis described above and present the results in Supplementary Table 6.

**Biophysical modelling**. Heat-related mortality is tightly linked to EWL in birds and hence is an accurate representation of the environmental stress experienced by a bird in its natural habitat[9]. We used an R implementation of an endotherm biophysical model in Niche Mapper, forced by the microclimate model of the R package NicheMapR (v1.3), to explore how plumage reflectivity affected daily EWL in birds. These models have been described and tested in detail elsewhere[36,37]. Briefly, the microclimate model takes daily weather and terrain data to compute hourly radiation (long and short wave), air temperature, wind speed and humidity at organism height for a particular location and solves a heat and water budget, taking into account biophysical attributes of an organism and its physiological and behavioural responses. Ultimately, it calculates the metabolic and EWL rates required to maintain a given core temperature for each hour, allowing the bird to vary its posture, skin thermal conductivity, body temperature, respiratory water loss and cutaneous water loss, in that order, under heat stress[38].

Given that the results of our comparative analysis revealed greater NIR reflectivity in arid environments and for small birds, we wanted to explore the magnitude of the fitness advantage that high NIR could provide in such environments. Since EWL is a measure tightly linked to mortality, we used this variable to quantify the effect that plumage reflectivity may have on a bird's fitness. To investigate this we chose a location that lies on the upper limit of the 95% confidence interval of our environmental PC2, located in the Simpson Desert Reserve in South Australia (−27.36, 138.71). We extracted environmental variables for this location for December, January and February (the hottest months in Australia) in 2016. Given that the effect of temperature may be different for different body sizes, and our comparative analysis showed that smaller birds have higher NIR, we ran the endotherm model for three different body sizes (10 g, 100 g and 1 kg), which correspond to the upper and lower 95% limits of the size distribution in our data set and a middle point. These body sizes were chosen to illustrate the effects of NIR variation on EWL for birds of different sizes. Since we were mainly interested in the effect of dorsal plumage reflectivity on EWL, for each size we ran the model using one of two different values of dorsal reflectivity (either 0.1 or 0.4). These reflectivity values correspond to the average NIR reflectivity in our data set, minus and plus two standard deviations, while keeping UV-visible reflectivity constant at the average value (0.2). We weighted these values according to the percentage of radiating sunlight in each part of the spectrum (49% for UV-visible and 51% for NIR). We varied dorsal reflectivity while keeping constant ventral reflectivity at the average level (at 0.4, see Supplementary Figure 7 for a schematic showing how NIR reflectivity values were derived for biophysical models).

We obtained information on climatic from continent-wide 0.05° grids of interpolated weather data produced by the AWAP database[29] and soil properties from the Soil and Landscape Grid of Australia[39]. In the model we assumed that birds were foraging on the ground, under full sun, which is common in arid environments. Because we were specifically interested in isolating the potential effect of NIR reflectivity, behavioural variables were held constant (i.e. we assumed no behavioural thermoregulation). Given that climatic conditions vary at different distances from the ground, we computed microclimatic conditions (wind speed, air temperature and humidity) at 5 cm from the ground for a 10 g bird, 10 cm for a 100 g and 20 cm for a 1 kg bird. To check that the benefit of higher NIR reflectivity in small birds was not due to closer proximilty to the ground, where air temperatures are higher and wind speeds lower, we modelled 10 g birds at the same foraging height as large birds (20 cm) and found a similar benefit (2.14% body mass per hour water loss less for high than low NIR reflectivity; Fig. 3a). The same parameters were used for the extraction of environmental variables in Tonto National Forest, Arizona, USA (33.329, −111.5538), but in this case the variables were extracted from the University of Idaho Gridded Surface Meteorological Data daily weather database (https://www.northwestknowledge.net/metdata). All environmental parameters are summarised in Supplementary Table 8.

Most of the biophysical parameters were based on Kearney et al.[38] model for night parrots, but we adjusted some of these proportionally to body size. We extracted values on feather length and feather depth from published literature[40–45] and used these values to generate estimates that matched the three body sizes we used for the models.

The equations used to derive feather depth and feather length were:

Depth (m): $-0.2082 \times mass^2 + 0.1339 \times mass + 0.001$, but depth = 0.02 for a mass of 1000 g.

Length (m): $0.0085 \times \log(mass) + 0.0526$.

Basal heat generation (*W*) was also adjusted according to weight (*m*, kg) using the equation $0.034594 m^{0.669}$ for birds[46]. All the other parameters were kept constant for all body sizes. All biophysical parameters are presented in Supplementary Table 8.

**Model validation**. To test how accurately the model described in the main text can predict the physiological response of a bird, we compared the results of the model with that of published results on rates of EWL as a function of air temperature (from 30 to 50 °C at 2 °C intervals) in a small desert bird, *Auriparus flaviceps*[44]. The fit of the model prediction to the empirical data is shown in Supplementary Figure 5. The empirical data are from experiments on live birds in a metabolic chamber so we compared fit of the data to a model where all environmental parameters but air temperature are held constant. In the model, values of feather length, depth and basal metabolism changed proportionally to body size.

The map of aridity in Australia shown in Fig. 3 was extracted from the Atlas of Living Australia (ALA) and represents a dimensionless indicator of the degree of dryness of the environment, which is calculated using the monthly ratio of precipitation to potential evaporation. This measure is strongly correlated with precipitation, temperature and solar irradiation. The points plotted in the figure represent the average latitude and longitude of 1000 random records of each species' distribution, downloaded from the ALA.

## Data availability

The data sets generated during the current study are available in the Figshare repository https://doi.org/10.6084/m9.figshare.6818813.v1.

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

## Acknowledgements

We thank John Endler, Naomi Langmore, Adnan Moussalli, Ben Phillips and Michael Jennions for discussion and comments. This work was supported by the Australian Research Council (DP120100105 to D.S.-F. and DP150101652 to R.A.M.), University of Melbourne McKenzie Fellowship to I.M. and an Australasian Society for the Study of Animal Behaviour grant to E.N.

## Author contributions

D.S.-F. conceived the study; all authors contributed to study design; E.N. collected the data; I.M. performed phylogenetic comparative analyses; I.M. and M.R.K. performed biophysical modelling analyses; D.S.-F. and I.M. wrote the manuscript draft; all authors contributed to editing the final paper.

## Additional information

**Competing interests:** The authors declare no competing interests.

