## [Peer Review File · Nature Communications]

Reviewers' comments:

Reviewer #1 (Remarks to the Author):

This is an elegant and important study, and I congratulate the authors for producing a body of work that will have far-reaching implications for our understanding of how birds interact with their thermal environments.

The only aspect of the manuscript I would query concerns the biophysical modeling, from which the authors conclude that higher reflectance of NIR among arid-zone species can translate into savings in evaporative water loss requirements of up to 2 % of body mass per hour in ~10 g species. The biophysical model used by the authors predicts that 10-g species lose > 10 % of body mass per hour as evaporative water loss (lines 107-109). There is no question that rates of water loss as a percentage of body mass do become very high when air temperature approaches 50 °C, but I would question whether they routinely reach as much 10 % body mass per hour. For instance, empirical measurements of maximum evaporative water loss for small birds at high air temperature suggest rates equivalent to approximately 5 % body mass per hour in 7-g Verdins exposed to air temperatures (T_a) around 48 °C (Wolf and Walsberg 1996 J. Exp. Biol.), 9 % in 10-g Lesser Goldfinches at $T_a = 50$ °C (Smith et al 2017 J. Exp. Biol), 4.3 % in 10-g Scaly-feathered Weavers at $T_a = 48$ °C (Whitfield et al 2015 J. Exp. Biol.) and 5 % in 17-g Yellow-plumed Honeyeaters at $T_a = 46$ °C (McKechnie et al 2017 J. Exp. Biol.). The picture that emerges from these empirical data is of maximum rates that are typically closer to 5 % of body mass per hour than 10 % - this is the reason I am wondering about the latter value and authors' estimated savings of around 2 % of body mass per hour. The above-mentioned studies involved exposing birds to the highest air temperatures they could tolerate before the onset of severe hyperthermia, and involved experimental conditions under which very low humidities were used to eliminate any impediment to evaporative cooling. So I am concerned that there may be a mismatch between observed maximum avian rates of evaporative water loss and the rates predicted by the biophysical model used by authors of the present manuscript.

Reviewer #2 (Remarks to the Author):

The authors use uv-vis-NIR spectrometry of museum specimens, and use them to build a biophysical model of heat transfer and to perform comparative analyses relating coloration to environmental variables. These data suggest that coloration may help buffer birds body temperatures, particularly against extreme temperature conditions. The paper is on an interesting topic that has received relatively little attention (the recent (and puzzlingly here-uncited paper) by Shawkey et al. 2017 on sunbird uv-vis-NIR reflectance is an exception) and is clearly written and compelling. However, it is also oversold and contains some puzzling inconsistencies and logically dubious lapses in data analysis. Specifically:

- 1) the authors focus most of their attention on analysis of NIR reflectance, with Uv-vis excluded controlled for (relative NIR). There is no reason to do this because all wavelengths have thermal properties. Just because a wavelength is not in the range of those we typically

associate with heat (perhaps bc we can't see it) does not mean that it has no effect on heat. So excluding wavelengths from 300-700nm from comparative analysis is not justified or needed. Comparing visible and NIR reflectance is interesting as a sidenote, but the full spectrum data should be used in all of the main analyses.

2) How do results change when combining data from all body parts rather than separately? It is interesting that trends are stronger in parts exposed more to the sun, but is there any difference in reflectance from the whole body?

3) It is interesting that the results from males and females considered separately are similar, but how do they compare in absolute terms? I.e. do females reflect have greater solar reflectivity than males? Does the bio physical model predict the same effects for males and females considered separately?

4) The biophysical model is interesting, and certainly suggests that color may be important at least in some extreme cases, but it also simplified and does not take into account other critical factors that may affect heat loss/gain (e.g. behavioral ones like panting, feather fluffing, differences in feather density or structure, the effect of wind). The only way to do this is through direct physiological data, which are sparse in the literature. Thus, the strong claims of the paper (e.g. in the title) should be toned down to more accurately reflect the results.

Specific comments:

L42: See specific comment 1) above. There is no need to control for UV-vis reflectance. Just analyze it along with the other wavelengths that have thermal properties.

L108: These seem like pretty extreme conditions that most birds do not experience. Is there any reason to think that this plumage color would be relevant for birds living in more typical climactic conditions?

Reviewer #3 (Remarks to the Author):

Review of Reflection of near-infrared light confers thermal protection in birds

The authors present an interesting comparative study examining the evolutionary diversity of near-infrared reflectance of numerous Australian bird species. They found that species in hot and dry environments reflect more energy in the near infra-red wavelengths than species in other environments. This effect was weaker in larger bird species.

This is an interesting study that should appeal to a broad array of ecologists and evolutionary biologists.

Most of the paper is well written, though I do have a couple of methodological questions.

Lines 217-218: What proportion of the total Australian avifauna does this represent?

Line 227: This paragraph would be better positioned as the first one in this section. In addition, the sentence starting on line 233 would fit better as the first sentence in this paragraph.

Line 268: It's not clear what data were used in the residual analyses. Were these the reflectance values averaged across body parts for each species? In addition, the regressions used to generate the residuals should also account for phylogeny. All of that being said, I would recommend against the use of residuals used as data in subsequent analysis. Freckleton summarizes these problems well in his 2009 paper. Alternatively, the use of visible reflectivity as a covariate is a better approach.

Lines 335-336: Do you have a references for using delta AIC >3?

Reference

Freckleton, R. P. "The seven deadly sins of comparative analysis." *Journal of evolutionary biology* 22, no. 7 (2009): 1367-1375.

We would like to thank the opportunity to re-submit our work. Please find attached our revised manuscript with a point-by-point response to all of the reviewers comments.

We have performed several additional analyses and included extra tables and figures in the supplementary material, following the reviewers suggestions. We believe that this version has greatly improved in quality and clarity thanks to their comments, and we hope it has made our study suitable for publication in *Nature Communications*.

Reviewers' comments:

Reviewer #1 (Remarks to the Author):

This is an elegant and important study, and I congratulate the authors for producing a body of work that will have far-reaching implications for our understanding of how birds interact with their thermal environments.

Response: We thank the reviewer for their positive appraisal and respond to specific comments below.

The only aspect of the manuscript I would query concerns the biophysical modeling, from which the authors conclude that higher reflectance of NIR among arid-zone species can translate into savings in evaporative water loss requirements of up to 2 % of body mass per hour in ~10 g species. The biophysical model used by the authors predicts that 10-g species lose > 10 % of body mass per hour as evaporative water loss (lines 107-109). There is no question that rates of water loss as a percentage of body mass do become very high when air temperature approaches 50 °C, but I would question whether they routinely reach as much 10 % body mass per hour. For instance, empirical measurements of maximum evaporative water loss for small birds at high air temperature suggest rates equivalent to approximately 5 % body mass per hour in 7-g Verdins exposed to air temperatures (T_a) around 48 °C (Wolf and Walsberg 1996 J. Exp. Biol.), 9 % in 10-g Lesser Goldfinches at $T_a = 50$ °C (Smith et al 2017 J. Exp. Biol.), 4.3 % in 10-g Scaly-feathered Weavers at $T_a = 48$ °C (Whitfield et al 2015 J. Exp. Biol.) and 5 % in 17-g Yellow-plumed Honeyeaters at $T_a = 46$ °C (McKechnie et al 2017 J. Exp. Biol.). The picture that emerges from these empirical data is of maximum rates that are typically closer to 5 % of body mass per hour than 10 % - this is the reason I am wondering about the latter value and authors' estimated savings of around 2 % of body mass per hour. The above-mentioned studies involved exposing birds to the highest air temperatures they could tolerate before the onset of severe hyperthermia, and involved experimental conditions under which very low humidities were used to eliminate any impediment to evaporative cooling. So I am concerned that there may be a mismatch between observed maximum avian rates of evaporative water loss and the rates predicted by the biophysical model used by authors of the present manuscript.

Response: The reviewer is correct that estimated evaporative water loss (EWL) from experimental studies in metabolic chambers is lower than the values reported by our model for birds in arid environments. The experimental values referred to by the reviewer are all derived from studies using metabolic chambers where there is no extra load from solar radiation. Our model predictions incorporate direct solar radiation, explaining the higher estimates.

To validate our model, we compared predictions of the model to experimental values from Wolf and Walsberg 1996 under equivalent conditions; i.e., with no solar radiation. Our model very accurately predicts empirical values from Wolf and Walsberg 1996 (Supplementary Figure S1). Thus, the higher estimates of evaporative water loss estimated by our model under natural environmental conditions (albeit extreme arid environments) are consistent with experimental data, and realistic. For example, the values in Albright et al. 2017 range between 8 and 10% body mass per hour evaporative water loss at 50C for a 10g goldfinch in a metabolic chamber; whereas our estimate for this species under exposed arid conditions is around 12% - slightly higher due to direct solar radiation, as expected. We now explain the discrepancy between previous estimates from metabolic chambers and estimates from our model (as well as how the model was validated) in the main manuscript (lines 121-125).

Reviewer #2 (Remarks to the Author):

The authors use uv-vis-NIR spectrometry of museum specimens, and use them to build a biophysical model of heat transfer and to perform comparative analyses relating coloration to environmental variables. These data suggest that coloration may help buffer birds body temperatures, particularly against extreme temperature conditions. The paper is on an interesting topic that has received relatively little attention (the recent (and puzzlingly here-uncited paper) by Shawkey et al. 2017 on sunbird uv-vis-NIR reflectance is an exception) and is clearly written and compelling. However, it is also oversold and contains some puzzling inconsistencies and logically dubious lapses in data analysis.

Response: We thank the reviewer for recognising the novelty and significance of our study. Shawkey et al describe variation in reflectance between colour patches, sexes and species of sunbirds and we now cite the study (we were aware of the study – one of us reviewed it – but could not cite it at the time of submission because it was not yet published). Shawkey et al – and other studies that have documented NIR reflectance in various taxa – could only speculate on its adaptive function. The significance of our study is that we show an evolutionary correlation with thermal environment and specifically model effects on a fitness-related trait (Evaporative Water Loss in arid conditions).

We address the reviewer's comments regarding data analysis and inferences below.

1) the authors focus most of their attention on analysis of NIR reflectance, with Uv-vis excluded controlled for (relative NIR). There is no reason to do this because all wavelengths have thermal properties. Just because a wavelength is not in the range of those we typically associate with heat (perhaps bc we can't see it) does not mean that it has no effect on heat. So excluding wavelengths from 300-700nm from comparative

analysis is not justified or needed. Comparing visible and NIR reflectance is interesting as a sidenote, but the full spectrum data should be used in all of the main analyses.

Response: We understand the reviewer's point and we use the full wavelength range in our biophysical models estimating effects of reflectance on evaporative water loss. However, we were specifically interested in whether selection for thermoregulation has shaped variation in NIR reflectance, independently of visible reflectance, which is strongly constrained by selection for optical functions (e.g. camouflage, communication). Comparison of drivers of visible and NIR reflectance variation provides a means to test whether animals have 'solved' the problem of competing selection for optical and thermal functions by modulating visible and NIR reflectance differently (e.g. low visible reflectance for camouflage offset by high relative NIR to mitigate risk of overheating). We would not be able to identify such patterns if we only examined correlations between total reflectance (UV+vis+NIR).

Nevertheless, we have run models using total reflectance as the response variable as suggested by the reviewer (now described in the main text – lines 89-95 – and presented in Supplementary Table S8). Results support our main conclusions: we find the same association between environment and reflectance for dorsal, but not for ventral body regions.

2) How do results change when combining data from all body parts rather than separately? It is interesting that trends are stronger in parts exposed more to the sun, but is there any difference in reflectance from the whole body?

Response: Results of the analysis when combining data from all body parts are similar to those for dorsal body regions, for which associations with environmental variables were stronger than for ventral body regions (and therefore drive patterns for all patches combined). For all patches combined, the best model in 89% of cases includes both environmental PCs and their interaction with size. The best model was on average more than 1000 times better than the null model. We now include this information in the main text (lines 85 to 88) and present full results of the analysis in Supplementary Table S6.

3) It is interesting that the results from males and females considered separately are similar, but how do they compare in absolute terms? I.e. do females reflect have greater solar reflectivity than males? Does the bio physical model predict the same effects for males and females considered separately?

Response: Below we present the values of NIR and visible reflectivity for males and females, including bars that represent standard deviations. There is a tendency for females to have higher reflectivity in ventral patches. We present this graph and associated analysis in the supplementary material:

Supp. Material: 'Overall we found that females tend to have higher values of NIR reflectivity and visible reflectivity compared to males, but only in the ventral patches (Figure S4). This is identical to what was recently found by Shawkey et al. in sunbirds, and probably responds to the density of feathers and insulation in these two patches. The difference between males and females could be due to the importance of these patches in incubation, but this remains speculative.'

Figure S4 (shown in supplementary material)

Regarding the biophysical model, we did not model our results separately for males and females because species vary in sexual size dimorphism and we used data on species average body size. Furthermore, differences between males and females are small relative to differences between species, such that they are unlikely to result in different patterns of evaporative water loss.

4) The biophysical model is interesting, and certainly suggests that color may be important at least in some extreme cases, but it also simplified and does not take into account other critical factors that may affect heat loss/gain (e.g. behavioral ones like panting, feather fluffing, differences in feather density or structure, the effect of wind). The only way to do this is through direct physiological data, which are sparse in the literature. Thus, the strong claims of the paper (e.g. in the title) should be toned down to more accurately reflect the results.

Response: We thank the reviewer for prompting us to clarify parameters in the biophysical model. The model does incorporate parameters related to the effect of wind, feather density, and feather length. These are shown in Table S9 in the supplementary material. The model also allows for behavioural and physiological responses, allowing the bird to vary its posture, skin thermal conductivity, body temperature, respiratory water loss and cutaneous water loss in that order under heat stress, which we now emphasise in the methods (lines 393 -395). We therefore do not feel that our claims are too strong but would be happy to modify the title if requested by the associate editor.

Specific comments:

L42: See specific comment 1) above. There is no need to control for UV-vis reflectance. Just analyze it along with the other wavelengths that have thermal properties.

Response: See response to Reviewer 2, comment 1 above. Different parts of the spectrum (Visible vs. NIR) are likely to be under different selective pressures (or trade-

offs between them), hence we would not expect them to be shaped in the same way by the environment. We have done the combined analysis as suggested but think that we would lose important information by not including analyses of visible and NIR wavelengths separately.

L108: These seem like pretty extreme conditions that most birds do not experience. Is there any reason to think that this plumage color would be relevant for birds living in more typical climactic conditions?

Response: We recognise that environmental parameters in our model represent 'extreme conditions', but almost half of the Australian continent reaches these extreme conditions during summer. The temperatures used in our model are highly relevant for at least 10% of the species of Australian passerines, which have a centre of their distribution within these arid regions. In fact, according to the Atlas of Living Australia, between 2001 and 2018, more than 100 species of passerines from 70 different genera were reported in the region of the Simpson Desert (the location used in our model) and there were more than 3000 records (see map below). To us, this suggests that these temperatures are actually experienced by a significant number of birds. Additionally, recent data indicates that the incidence of heat waves and extreme high temperatures will increase (1) and that arid and hyper-arid zones will likely expand (2). This means that the likelihood of birds encountering these conditions will probably increase in the future. We now include an additional paragraph explaining why our results are likely relevant to a significant proportion (albeit a minority) of bird species (lines 144– 153). Of course, plumage colour is likely to be less relevant in terms of thermal benefits for species living in more benign conditions, which we clearly state (lines 169 – 170).

Editorial Note: Map courtesy of the Atlas of Living Australia, available under a CC BY 3.0 AU license.

References:

1. Coumou, D., & Robinson, A. (2013). Historic and future increase in the global land area affected by monthly heat extremes. *Environmental Research Letters*, 8(3), 034018.
2. Zarch, M. A. A., Sivakumar, B., Malekinezhad, H., & Sharma, A. (2017). Future aridity under conditions of global climate change. *Journal of Hydrology*, 554, 451-469.

Reviewer #3 (Remarks to the Author):

Review of Reflection of near-infrared light confers thermal protection in birds

The authors present an interesting comparative study examining the evolutionary diversity of near-infrared reflectance of numerous Australian bird species. They found that species in hot and dry environments reflect more energy in the near infra-red wavelengths than species in other environments. This effect was weaker in larger bird species.

This is an interesting study that should appeal to a broad array of ecologists and evolutionary biologists.

Most of the paper is well written, though I do have a couple of methodological questions.

Response: We thank the reviewer for recognising the significance and broad appeal of our study. Below we reply to the specific methodological questions.

Lines 217-218: What proportion of the total Australian avifauna does this represent?

Response: It represents 12% of the species, 35% of the families and 66% of the orders of Australian birds, we have now added this information to the manuscript. Line 44.

Line 227: This paragraph would be better positioned as the first one in this section. In addition, the sentence starting on line 233 would fit better as the first sentence in this paragraph.

Response: We have moved this paragraph and changed the first sentence as suggested.

Line 268: It's not clear what data were used in the residual analyses. Were these the reflectance values averaged across body parts for each species? In addition, the regressions used to generate the residuals should also account for phylogeny. All of that being said, I would recommend against the use of residuals used as data in subsequent analysis. Freckleton summarizes these problems well in his 2009 paper. Alternatively, the use of visible reflectivity as a covariate is a better approach.

Response: Following the reviewer's suggestion we have re-calculated the residuals using a phylogenetically controlled model and we have re-run all comparative analyses. We have updated data tables with the new values. Patterns remain qualitatively identical.

Also, we understand the potential problems with using residuals, which is why we also used initially the reviewer's suggested approach of including visible reflectivity as a covariate (Extended data in the original submission). However, we now agree with the reviewer that it would be better to present results with visible reflectivity as a covariate

in the main manuscript, which we now do. We have updated figures and methods accordingly. We continue to show the analysis of residuals but as an alternative approach in the supplementary materials.

Lines 335-336: Do you have reference for using delta AIC >3?

Response: The cut-off value of three (or two or four) is arbitrary, and there is still debate about when a model can confidently be considered uninformative (Symonds and Mousalli 2011) but three has been used in several publications (e.g. Lessard et al. 2014, Rowley & Alford 2013), and we have added a reference to it in the main text (line 363). Nevertheless, we understand the point of the reviewer and have confirmed that our results and conclusions remain identical if we use a cut-off value of two, three or four.

Symonds, M. R., & Moussalli, A. (2011). A brief guide to model selection, multimodel inference and model averaging in behavioural ecology using Akaike's information criterion. *Behavioral Ecology and Sociobiology*, *65*(1), 13-21.

Lessard, E., Fournier, R. A., Luther, J. E., Mazerolle, M. J., & Van Lier, O. R. (2014). Modeling wood fiber attributes using forest inventory and environmental data for Newfoundland's boreal forest. *Forest ecology and management*, *313*, 307-318.

Rowley, J. J., & Alford, R. A. (2013). Hot bodies protect amphibians against chytrid infection in nature. *Scientific Reports*, *3*, 1515.

REVIEWERS' COMMENTS:

Reviewer #1 (Remarks to the Author):

The authors have adequately addressed the concerns I raised in my original review. I congratulate them on an important piece of work.

Reviewer #2 (Remarks to the Author):

The authors have done a nice job addressing my comments, and I have nothing further to add.

Reviewer #3 (Remarks to the Author):

The authors have done a good job replying to the reviewers' comments. I look forward to seeing the paper published.